# Precarious Suicide Behavior According to Housing Price Gap: A Case Study on South Korea

**DOI:** 10.3390/ijerph18189877

**Published:** 2021-09-19

**Authors:** Sungik Kang, Hosung Woo, Ja-Hoon Koo

**Affiliations:** Department of Urban and Regional Development, Hanyang University, Seoul 04763, Korea; namugnel@gmail.com (S.K.); woo2811@gmail.com (H.W.)

**Keywords:** suicidal impulse, housing price gap, housing price inequality, multilevel logit model, South Korea

## Abstract

In 2018, the suicide rate in South Korea was the highest among the Organisation for Economic Co-operation and Development countries, and socioeconomic inequality has intensified. This study analyzes the impact relationship between suicidal impulses and economic inequality in South Korea. This study measures suicidal impulses thoughts National Health Survey Data and economic inequality based on the housing prices gap in the country. The primary analysis results were as follows: First, suicidal impulses were positively associated with the high index of housing price inequality; this correlation has become tight in recent years. Second, it was confirmed that the higher the income level, the higher the correlation between suicidal impulses with the index of housing price inequality. Third, the correlation between housing price inequality with suicidal impulse increased consistently in highly urbanized areas, but the statistical significance was low in non-urban areas.

## 1. Introduction

According to World Health Organization (WHO) statistics, globally, approximately 788,000 people die from suicide each year, and suicide is a significant health issue worldwide [1]. Among all the countries in the world, the suicide rate in South Korea is exceptionally high. The South Korean suicide rate is the highest among the Organisation for Economic Co-operation and Development (OECD) countries, more than twice the OECD average of 10.9% [2]. According to the 2018 (OECD) statistics, in South Korea, 23 people per 100,000 people committed suicide; the suicide rate in South Korea has been the highest among the OECD countries for 10 years. In South Korea, suicide is now the most common cause of death for the age group of 10–39 years and the second most common cause of death for the age group of 40–59 years [3]. About 3% of South Korean adults over the age of 18 said they had suicidal thoughts, and about 2.4% said they had attempted suicide [4]. As described by various statistics, suicide in Korea is a severe social problem regardless of age [5].

In South Korea, along with suicide, there are concerns and discussions about economic inequality. In South Korea, the problem of socioeconomic inequality has not emerged as a significant social issue during the industrial growth period in the 1970s and 1980s. However, since the middle of the 1990s, the socioeconomic inequality of South Korea has emerged as a side effect of tremendous economic growth. The Gini coefficient of South Korea in 2017 was 0.36, which indicates inequality in the income of the society [6,7]. South Korea’s Gini coefficient shows income inequality close to that of the United States, the highest among the OECD countries; this accelerating inequality is recognized as one of South Korea’s most serious social problems today [7]. As of 2018, South Korea ranked 34th in the top 10% and bottom 10% wage comparison index (interdecile P90/P10) among 37 OECD countries [8]. Recently, the inequality index in the country has been worsening [7].

Academia argues that suicidal behavior and increasing inequality are not separate phenomena, belonging to the same social stream. Many studies have hypothesized that suicide and income inequality are social issues, with a reasonably close relationship [9,10,11,12,13,14]. This theory has developed owing to the view that the influence of individual income level decreases for health; however, the effect of regional income gaps has become critical [15,16]. Wilkinson (2005) suggested that increasing income inequality forms a mechanism that widens social status gaps, worsens social trust levels and leads to depression and insecurity. Several studies have demonstrated statistically significant correlations between income inequality and suicide rates in specific cohorts [10,17,18].

Western academia has principally studied inequality and suicide [19]; however, studies on South Korea, which has the highest suicide rate among the OECD countries, are still insufficient. Notably, previous studies on income inequality and suicide rates were mainly empirical. However, this study attempts to analyze the association between suicidal behavior and housing asset inequality based on housing prices. While rapidly increasing housing prices and housing price increases due to housing speculation may cause depression, this study focused on the correlation with suicidal behavior according to the increase in the housing price gap. In addition, we analyzed the recent trends in correlations, differences in correlations by income level and regional characteristics. The housing price inequality variable used the Ministry of Land, Infrastructure and Transport’s (MOLIT) housing transaction price data, and the suicide behavior variable used data from the National Health Impact Survey. This study analyzes the relationship between unequal social environment and suicide in South Korea. We analyzed the data using a three-level multilevel logit model employing regional and individual unit data over a decade (2008–2018).

## 2. Literature Review

### 2.1. Hypothesis and Theory-Related Income Inequality and Suicide

It has been hypothesized that income inequality is strongly related to the quality of social relationships and personal health [19,20]. Several studies have shown that lower income inequality is more beneficial to health [21]. Many studies have also explained the mechanism by which high-income inequality arouses social comparison, relative deprivation and the displeasure of injustice, and consequently leads to mental stress and suicide [22,23]. For example, as income inequality worsens, the consumption of the top class becomes the standard, and the relative sense of deprivation and depression of the middle and lower classes may increase [24]. Income inequality stimulates these negative emotions and acts as a factor in encouraging suicidal behavior [23]. In addition, research on social capital has demonstrated that an increase in community bonds positively affects the health of the population [25]. In particular, studies have explained that social capital and suicide have a negative correlation, and high social capital is associated with a lower suicide rate [26,27]. Durkheim studied the increased probability of suicide with low social integration, and Congdon developed Dukheim’s research to emphasize the increased suicide rate in societies having social fragmentation [19,28,29].

Previous studies have hypothesized that income inequality and suicide are closely related [9,10,11,12,13,14], and various related studies have also been conducted. Sociologists, including Emile Durkheim, have predicted that social structures and local pathology had highly associated with suicide [29,30,31]. In social disorganization theory, researchers have determined that there is an essential connection between income inequality and suicide with respect to urbanization, family breakdown and low social status [32,33]. Other studies have also analyzed that social pathologies, such as poverty, unemployment and social fragmentation are associated with suicide [34,35], although empirical evidence is mixed and controversial [20].

### 2.2. Empirical Research on Income Inequality and Suicide

Among the existing studies on income inequality and suicide, the study findings in western countries are as follows: In a study of 20 developed countries worldwide (1955–1989), high-income inequality at age 45 and older was associated with high suicide rates [17]. In 12 countries in central and eastern Europe (1990–1993), perceptions of income inequality and suicide rates had a consistent correlation in men rather than women [10]. In a study of 15 European countries (1970–1998), there was also a positive correlation (r = 0.296) between male suicide rates and the Gini coefficients [20]. In a study of adolescent suicide in 11 countries worldwide, adolescent suicide had a statistically significant correlation, with high gross domestic product (GDP; Wilcoxon rank-sum test z = −2.27) and high-income inequality (Wilcoxon rank-sum test z = −2.45). In addition, in Australia, Canada and the United States of America, an increase in income inequality is associated with an increase in adolescent suicide rates (Wilcoxon rank-sum test, z = −0.29) [18].

The results of research on income inequality and suicide rates in Latin America and Asia are as follows: In a study using Brazilian panel data (2000–2011), the income Gini index was positively correlated with the male (rate ratio=1.060) and female (rate ratio=1.056) suicide rates [23]. A study in Japan also found that high-income inequality and high suicide rates were statistically related [36]. However, in a study analyzing income inequality and suicide rates in Taiwan (2004–2010), There was no clear correlation between income inequality with suicide at basic models, model by gender and model by age [19]. A study in South Korea (2012–2018) found that an economically low social status showed a positive correlation with suicidal thoughts and attempts [5]. Another suicide study in South Korea (1995–2005) analyzed that the relative inequality index generally increased in males and the absolute inequality index generally increased in females [37].

Among the existing studies on suicide rates and income inequality, the results of a specific cohort are as follows: In a study of New York City in 1996, the suicide rate was high in villages, with high-income inequality observed for the age group of 15–34 years. The results were similar for the age group of 35–64 years, although the statistical significance was low [13]. Suicide in men aged 25–34 in England and Wales (1950–1998) has been statistically associated with divorce, declining marriages and high-income inequality [11]. A study in New Zealand (1991–1994) found that a 0.01% increase in income inequality increased the suicide rate in men by 2.1% and women by 8.7%, although the 95% confidence interval included 1.0 [9].

### 2.3. Other Findings

Several studies have demonstrated associations of wealth disparities, such as homeownership and car access, with suicide [38]. For example, in a Denmark study, males with lower quartile wealth were more likely to commit suicide than upper quartile wealth [39]. A study of 10 European countries showed that suicidal behavior had a stronger correlation with housing ownership than education level [38]. A study in England and Wales found that people who were housing renters or had less car access were more likely to commit suicide [40]. A study analyzing New Zealand census records found that fewer cars were more likely to commit suicide [41].

However, several previous studies failed to prove a correlation between income inequality and suicide rate or had a negative correlation. In a study of 21 developed countries, income inequality and suicide rates were not statistically significant [42]. In a New Zealand study, the statistical significance between income inequality and suicide rates was not clearly demonstrated [9]. In a study of 3108 counties in 1039 rural areas across 50 US states, income inequality and suicide were of low statistical significance [43]. In contrast with previous studies, in a study of 88 counties, income inequality and suicide were negatively correlated [44]. As a result, the correlation between income inequality and suicide rate differed depending on the social characteristics and context of the region.

### 2.4. Our Contribution

First, our study examines the correlation between housing asset inequality and suicide based on housing prices, rather than income inequality. Notably, existing studies mainly analyze income inequality, whereas our study focuses on housing asset inequality. Assets are long-term economic indicators that differ in income and characteristics [45], and income inequality and asset inequality need to be analyzed separately when studying inequality [46]. Therefore, we investigated the correlation between housing asset inequality with suicide, focusing on housing prices. Second, to illuminate the relationship between housing asset inequality and suicide rate, we analyzed the characteristics of recent correlations. Third, we examined how the association between housing asset inequality with suicidal impulses varies according to income level. Fourth, we divided the regions into capital and non-capital areas, metropolitan and non-metropolitan areas, and urban and non-urban areas and analyzed the difference in correlation between housing price inequality and suicide, according to regional characteristics.

## 3. Research Framework

### 3.1. Suicide Rate and Housing Price Inequality in South Korea

As explained in the OECD statistics above, the suicide rate in South Korea is very high. As of 2019, the suicide rate by region and gender per 100,000 people in South Korea was the highest in Chungcheongnam-do (35.0) and lowest in Sejong City (21.4; Figure 1). The suicide rate by gender was even more dramatic. The national suicide rate for men was 37.6 and 15.7 for women, a difference of about 2.4 times. The male suicide rate, even though lowest at 25.3 in Sejong City, is the highest among the OECD countries; and it is very high at 50.3 in Chungcheongnam-do. Additionally, in female suicide rates, the rate was highest in Chungcheongnam-do (19.1) and lowest in Jeollanam-do (11.3; Figure 1). The male suicide rate is well above the average of OECD countries, and the female suicide rate is slightly above the average of OECD countries.

The second (b) of Figure 2 shows the housing price inequality index in 2018, the most recent among the study’s periods. In 2018, the housing price inequality index of South Korea was 0.336, and Seoul (0.271) was the highest inequality region. On the other hand, the region with the lowest housing price inequality index is Gwangju (0.208), and the difference between Seoul and Gwangju housing price inequality index is more than 0.06. The change rate between 2013 and 2018, which has the most change, is shown in the third (c) of Figure 2. The change rate of South Korea’s housing price inequality index rose to 12.9%. As for the change rate of housing price inequality index by regions, Gyeonggi-do showed the most increase (18.2%; 0.204→0.242), followed by Busan (15.6%; 0.206→0.239). In both regions, the rise was steep, although the current housing price inequality level is not high.

### 3.2. Data

This study used data from the National Health and Nutrition Survey of the Ministry of Health and Welfare, Centers for Disease Control and Prevention, and the housing transaction price data of MOLIT, which are representative statistical data of the country. The dependent variable (suicidal behavior) was used in the National Health and Nutrition Survey, a national health statistics survey that measures over 500 health conditions, such as health, nutrition and personal characteristics. It was introduced in 1998 and annually surveys 10,000 people. This dataset has a system that enables comparisons between countries by examining the health indicators required by the OECD and WHO. Our study established the suicidal impulse variable using the questionnaire item “suicide thoughts and attempts in one year” of the National Health and Nutrition Survey. If responses were choosing at least one in the two suicide-related items (suicidal thoughts and suicide attempts) for one year, we considered the respondents to have felt suicidal impulses. Other primary health states, such as physical and mental health, were included as control variables. In addition, we employed crucial items, such as gender, age, income, educational background, marital status and number of family members, in this study.

The housing price inequality index was measured using the housing transaction price of the MOLIT. The MOLIT has been disclosing all housing transaction prices nationwide to the public since 2006, allowing precise measurement of the housing price inequality index. The types of housing that are open to the public include all residential housing, such as apartments and multi-family and single-family housing. In this study, housing price inequality indexes for 17 provinces were estimated using housing transaction prices. Even though there are various methods of measuring inequality, we measured the degree of inequality using the Gini coefficient, which covers all distributions, not just the two tails of the group, and is most commonly employed in similar and related studies [47,48].

### 3.3. Method

In this study, we utilized a three-level multilevel model as the research methodology. The data in this study are composed of individual variables (at level 1) belonging to a specific region (at level 2) in a specific year (at level 3), and thus, have the characteristics of three-level multilevel data. In addition, because the dependent variable (suicide behavior) was measured as a binary variable (yes/no), we employed a three-level multilevel logit model. The equation of the model is as follows [49,50].
(1)$$g\left(\pi_{ijt}\right)=\log\left(\frac{\pi_{ijt}}{1-\pi_{ijt}}\right)=\alpha +Z_{ijt}\delta +\beta X_{ijt}+v_{t}+u_{jt}+e_{ijt}\;v_{t}\sim N\left(0,\;\sigma_{v}^{2}\right),\;u_{jt}\sim N\left(0,\;\sigma_{u}^{2}\right),\;e_{ijt}\sim N\left(0,\;\sigma_{e}^{2}\right),$$

In this formula, g($\pi_{ijt}$) consists of *i* respondents in area *j* of year t. $X_{ijt}$ is each independent variable, and β is the coefficient of each independent variable. $Z_{ijt}$ is the control variable, and δ is the coefficient value of each control variable. $v_{t}$, $u_{jt}$ and $e_{ijt}$ are the time effect of 10 years, regional effect of area *j* in year *t*, and residual error, respectively. We used a likelihood ratio (LR) statistical test that calculates the difference between the single-level and multilevel models to verify the multilevel effect in the model. In addition, we calculated the intraclass correlation coefficients (ICCs), which are the proportion of variance of the local housing price inequality index in the overall model.

## 4. Empirical Results

### 4.1. Descriptive Statistics

The basic statistics of the dependent and independent variables are listed in Table 1. Excluding 2014 which was not surveyed suicide-related questionnaires, the total number of samples in the National Health Impact Survey for 2008–2018 was 24,392. The rate of suicidal impulse among the respondents was 3.1%, and the standard deviation was 0.173. Housing price inequality is the Gini coefficient of 17 provinces over 10 years, with a maximum value of 0.352 (Jeollanam-do in 2017) and a minimum of 0.160 (Incheon, 2009). In terms of gender, females accounted for 59% and males accounted for 41%. Even though the raw data ratio of males was 45.3%, due to the high number of non-responses, the male ratio slightly decreased to 41.0%. The average age was 46.7 years, minimum was 12 years and maximum was 80 years. The proportion of married people is 26.7%, and the family with four members is the most common (27.8%). Regarding education level, high school graduation is the highest at 28.4%, and the annual average income is 30.9 million won. Among self-rated health, the mean of physical health was 3.09 points above median, and the mean of mental health was 2.75, which is above median.

### 4.2. Analysis Result of Relation between Housing Price Inequality and Suicidal Impulse

A study that analyzed the relationship between the 10-year housing price inequality index and the suicidal impulse in South Korea is as follows (Models 1, 2 and 3 in Table 2). The Akaike information criterion (AIC) and Bayesian information criterion (BIC) decreased as the model added explanatory and inequality variables; thus, the suitability of the analysis model increased. Notably, the correlation between suicidal impulses and housing price inequality was statistically significant. The odds ratio was 1.664 when the housing price inequality variable was the only parameter applied (Model 2) and 1.946 when the total independent variables were applied (Model 3). In other words, according to Model 3, which portrays the relationship between the two variables for 10 years, regions where a 0.1 high home price inequality index had 1.9 times higher suicidal impulses. The ICC of the second and third levels of Model 3, where total variables were applied, was 0.108, accounting for 10.8% of the variance in the housing price inequality index in the entire model.

Notably, the coefficient of correlation between the housing price inequality index and suicidal impulse has increased significantly over the past five years (Models 4, 5 and 6 in Table 2). As a result of the analysis, when only the housing price inequality index and suicide impulse variable were applied (Model 5), the coefficient value was 7.729. The coefficient value was 5.355 when all independent variables were applied (Model 6). In other words, regions where a 0.1 high home price inequality index had 5.3 times higher suicidal impulses. Compared to the 10-year coefficient of 1.946 between the housing price inequality index and suicidal impulse, the five-year coefficient increased to 5.355 (increase of 3.409). Thus, we can deduce that the correlation between the housing price inequality index and suicide impulse has risen significantly in the last five years.

The results of analyzing the correlation of housing price inequality according to demographic characteristics and health status are as follows (Model 3 in Table 2). Among the variables of demographic and sociological characteristics, suicidal impulses and statistically significant variables were gender, number of family members, education and income of the respondents. In contrast, age and marital status were not statistically significant. Notably, male suicidal impulses were 14.1% higher than those of females, and the fewer the family members, the higher the suicidal impulse (6.1%). We found that as the education level increased by one unit, the suicidal impulse decreased by 25.2%; an increase in the income by one unit resulted in a decrease in the suicidal impulse by 9.9%. Regarding the correlation between health status and suicidal impulse, when subjective health status increased by one level, suicidal impulse decreased by 24.1%. Finally, a higher level of mental health reduced suicidal impulses by 61.3%. As a result, both physical and mental health showed a close relationship with suicidal impulses.

### 4.3. Analysis Result by Income Level

Previous studies have found that in countries with increased per capita national income, the relationship between health with regional economic gaps is higher than individual economic levels [15,16]. Chiang (1999) found that as the region’s economic status improved during the study period, the income gap was more directly related to the mortality rate than the income level. In this study, we attempted to test the results of the above study according to income level by applying suicide impulses. Through this analysis, we observed how the statistical value of suicidal impulses changes according to the individual’s economic.

Our analysis classified the income class into the lower 20%, median 20% and upper 20% categories. The statistical results of the housing price inequality index and individual income level (by income class) are shown in Table 3. The odds ratio between the housing price inequality index with suicidal impulse were 3.793, 3.155 and 1.901 at the top 20%, median 20% and bottom 20% of income, respectively. The correlation between personal income and suicidal impulse was statistically significant only in the bottom 20%, with an odds ratio of 0.841 In other words, in the top 20%, while correlation suicidal impulses with total income are statistically insignificant, the odds ratio between suicidal impulses with the housing price inequality index is 3.793, which is considerably higher than that of other income classes. In the lower 20%, the suicidal impulse had a statistically significant correlation only with individual income level, and the odds ratio with the housing price inequality index was 1.901, which was lower than that of other classes. In summary, the relationship between individual income and suicidal impulse became lower in the high-income class, and the relationship between the housing price inequality index and suicidal impulse became higher in the high-income status. In other words, as the income level of individuals increased, the regional economic gap became more closely related to suicidal impulses than the individual’s economic level.

### 4.4. Analysis Result by Region Features

The results of the analysis based on regional location characteristics are shown in Table 4. The comparison of the Seoul-Gyeonggi area, which is the capital area of South Korea, and other areas is as follows (Model 1). In Seoul-Gyeonggi Province, suicidal impulses are 14.7 times higher in areas with a 0.1 high housing price inequality index and 1.7 times higher in non-Seoul Gyeonggi Province. In other words, people living in a place with a high inequality index in Seoul and Gyeonggi had much more suicidal impulses than those living in a different area, even if the demographic and sociological characteristics were the same.

The results comparing the seven metropolitan cities excluding Seoul Gyeonggi Province in South Korea, such as Busan, Daegu, Gwangju, Daejeon, Sejong and Ulsan, and the surrounding eight non-metropolitan regions are as follows (Model 2). Suicidal impulses in areas with a higher housing price inequality index of 0.1 are 3.1 times higher in metropolitan cities and 2.7 times higher in non-metropolitan regions. Even though there was no significant difference between the two regions, suicidal impulses in metropolitan cities were more sensitive to housing price inequality than those in non-metropolitan regions. In a comparative analysis of urban and non-urban areas (Model 3), suicidal impulses were 2.4 times higher and statistically significant in areas with a 0.1 high housing price inequality index in urban areas. In contrast, non-urban regions were not statistically significant. In other words, in urban areas, the suicidal impulse was high in areas with high housing price inequality. However, in non-urban areas, housing price inequality and suicidal thoughts had no significant correlation.

## 5. Discussion

### 5.1. Discussion within Significant Findings

This study examines the relationship between housing price gap and suicidal impulse in South Korea, which has high socioeconomic inequality and suicide rate. We employed suicidal thoughts and attempts from the National Health and Nutrition Survey data for suicidal impulses and measured the Gini coefficient using the housing prices of the MOLIT. Since the analysis data are composed of year, region and individual, this study used a three-level multilevel logit model as an analysis methodology. The main results and contributions of this study are as follows.

First, the correlation between suicidal impulse with the housing price inequality index has a close positive correlation, similar to income inequality, and this correlation has been becomingcloser in recent years. Similar to previous studies that demonstrate that the more significant the income gap, the higher the probability of suicide [9,10,11,12,13,14], this study confirmed that inequality in housing prices also correlates with suicidal impulses (OR = 1.946). The results of this study, similar to other previous studies [22,23], showed that negative emotions such as relative deprivation and suicidal thoughts were more felt in regions with a high economic inequality index. In addition, when comparing the correlation between 2008 and 2013, the odds ratio between the two variables increased significantly, with odds ratios of 1.946 and 5.355, respectively. This result supports the findings [37] that the correlation between income inequality and suicide is increasing over time.

Second, this study verified the phenomenon that the importance of individual income level decreases and the importance of regional income gap increases as personal income level increases about health level [15,16,51,52]. Existing Taiwanese studies have confirmed that as the economic level of residents increases, the death rate is more associated with regional income inequality than individual income levels [16]. While the Taiwanese study proved the phenomenon by the changes of the times, this study analyzed the phenomenon by the income classes. The results of this study are as follows: In the top 20%, the correlation between income level and suicidal impulse was found to be statistically insignificant. The odds ratio between housing price inequality and suicidal impulse was 3.793, which was significantly higher than that of the other income groups. In the lower 20%, the correlation between income level and suicide impulse was statistically significant. The odds ratio between the housing price inequality index and suicidal impulse was 1.901, which was significantly lower than that of the other income groups. Similar to the Easterlin paradox (1974), the results of this study showed that the influence on income decreased above a certain level of income, whereas the influence of the income gap increased [53]. As a result, our study proved that Pickett and Wilkinson’s (2010) research on the relationship between health and income level is insignificant, and the relationship between health and income inequality increases in more affluent countries is also applied to mental health (suicidal impulse).

Third, the correlation between housing price inequality and suicidal impulses differed depending on the degree of urbanization. In this study, we examined the correlation between housing price inequality and suicidal impulse according to the degree of urbanization (e.g., capital and non-capital areas, metropolitan and non-metropolitan regions, and urban and non-urban areas). The region where the correlation between housing price inequality and suicide impulse was the most negative was the Seoul-Gyeonggi region. The area with 0.1 high price inequality had 14.7 times higher suicidal impulse. The correlation between housing price inequality and suicidal impulse in metropolitan cities and surrounding autonomous cities was 3.136 and 2.753, respectively, proving that metropolitan cities showed a stronger negative relationship than general autonomous cities. In addition, the odds ratio of suicide impulse and housing price inequality in urban areas was 2.436, though statistically insignificant in rural areas. The results of this study were similar to those of research on income inequality, which explained that, in urban areas, suicide and economic inequality show significant correlations [43,54]. In addition, while previous studies examined the relationship between urbanization and suicide rates [5,19,23,37], this study analyzed the correlation between urban environment inequality and suicidal impulse according to the degree of urbanization.

### 5.2. Policy Implications

The policy implications of our study are as follows. First, we assumed that an increase in the housing price gap is statistically related to suicidal impulses. The study results suggest that the income gap among individuals and the gap in housing prices have a clear relationship with suicide. Therefore, not only the income inequality index but also the housing price inequality index need to be continuously monitored and discussed in diagnosing and solving various social problems, such as suicide. Since MOLIT publishes monthly housing prices, a system is required to determine which regions have higher housing price inequality indexes and which regions have increased housing price inequality indexes.

Second, South Korea needs to recognize that regional economic gaps are more important factors related to suicidal impulses than individual economic levels. In Korea, although the per capita national income was over $ 30,000 and the absolute income level of the people was very high compared to the past, the income inequality index (interdecile ratio p50/p10) is relatively high at 34th of 37 OECD countries. Our analysis confirmed that even though the current income is high, people living in an environment with a significant economic gap are more likely to have suicidal thoughts. South Korea’s income level is now much higher than before, reducing suicide concerns associated with low-income levels. However, South Korea is now time to worry about suicide concerns related to widening regional inequality, such as income inequality and housing asset inequality.

Third, the analysis results of our study consistently showed that the correlation between housing price inequality and suicidal thoughts was more significant in the positive direction in highly urbanized areas than in non-urban areas. In non-urban areas, the correlation between price inequality and suicidal thoughts was not statistically significant, and the correlation between the two variables was the most significant in the Seoul-Gyeonggi area. The Seoul-Gyeonggi region had the highest housing price inequality index, and the number of absolute suicide deaths was the highest in the research period. Therefore, monitoring and prevention activities related to suicide in areas with a large housing price gap (among the regions in Seoul-Gyeonggi) are considered to be effective in controlling the region’s suicide rate.

### 5.3. Research Limitations

The limitations of this study are as follows. First, we did not verify whether the 17 provinces (states) selected in the study were appropriate for measuring the local housing price inequality index. In other words, it has not been verified which variable among the inequality indexes at the national, state and county levels is more suitable for this study. Second, due to the limitation of statistical data, the study was conducted not on the same persons every year, but on other randomly selected respondents. Third, external socioeconomic factors, such as national growth rate and employment rate, that would affect individuals’ suicidal thoughts were not considered. Therefore, to obtain more detailed socioeconomic insights, a follow-up study is needed that considers other variables that affect suicidal thoughts, such as gross regional domestic product (GRDP), growth rate and employment rate. In addition, it is also necessary to study how the rate of change in housing price inequality correlates with suicidal ideation.

## 6. Conclusions

The main findings of this study are as follows. First, in areas with a high housing price inequality index, the probability of suicidal impulses was high, and the correlation between the two variables has recently become tighter. The index of home price inequality was 0.1, the probability of suicide was 1.9 times higher, and the possibility in the last five years has increased by 5.3 times. Second, the higher the income level, the correlation between suicidal impulses with regional economic gaps has increased; the lower the correlation between suicidal impulses with individuals’ economic levels. The odds ratio of correlation between the suicidal impulse with housing price inequality index was 1.90 times, 3.15 times and 3.79 times in the lower 20%, median 20% and upper 20%, respectively. Furthermore, the suicidal impulse and individual income were statistically significant only in the lower 20%. Finally, the correlation between the housing price inequality index and suicidal impulse was higher in urbanized areas, and the correlation in rural areas was not statistically significant.

The contributions of this study to previous studies are as follows: We verified that an unequal urban environment with a large housing price gap, similar to income inequality, correlates with suicidal thoughts. In addition, the results confirmed that the correlation between the two variables became closer over time. Second, the results of this study support the research results that as individual income levels increase, the importance of personal income levels decreases, and the importance of regional income gaps increases. This result revealed that both health-related problems that were dealt with in previous studies and mental health, such as suicidal impulse, illustrated a similar correlation. Third, the correlation between the two variables was that the correlation increases in the more urbanized areas and decreases in the rural areas.

The main implications of this study are as follows. In addition to income inequality, housing price inequality requires continuous monitoring. In an unequal urban environment, the increased tendency of suicidal thoughts has been confirmed. Therefore, the monitoring of areas having significant or severely increasing housing price gaps is required to prevent suicide. Second, countries with a certain level of national income, such as South Korea, should now consider the relative income gap on a regional scale, rather than improving the absolute income of individuals, to prevent suicidal thoughts. Therefore, while South Korea is less concerned about suicide impulses related to its low economic level and poverty, it is essential to keep in mind the suicidal impulse associated with the deepening economic gap. Third, to effectively prevent suicide, it is crucial to carry out suicide-related monitoring and prevention activities in areas with a large housing price gap among regions with a high suicide rate. In Korea, the suicide rate in the Seoul-Gyeonggi region is high, and the correlation coefficient between the housing price inequality index and suicidal thoughts is the largest. Thus, conducting preventive activities targeting unequal areas in the Seoul-Gyeonggi region is considered an effective way to lower the suicide rate.

## Figures and Tables

**Figure 1 ijerph-18-09877-f001:**
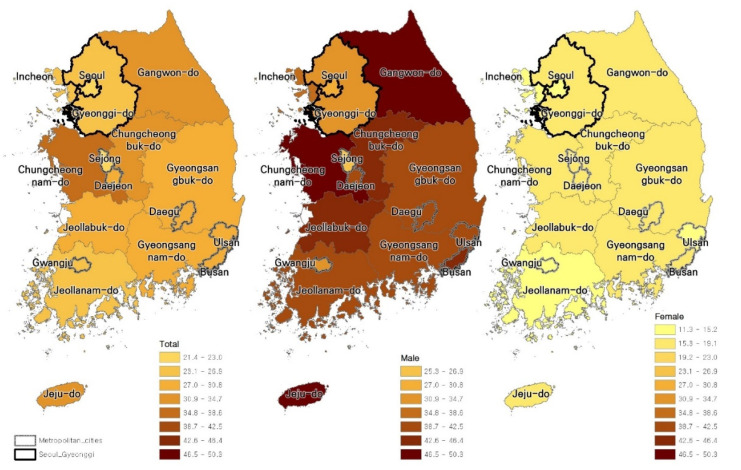
Suicide rate in South Korea for 2019, based on gender.

**Figure 2 ijerph-18-09877-f002:**
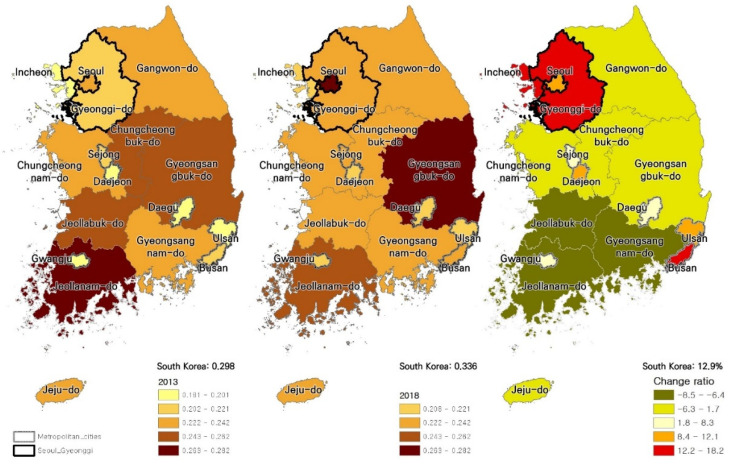
Housing Price Inequality Index and change rate in South Korea for 2013/2018. (a) Housing Price Inequality Index for 2013 (b) Housing Price Inequality Index for 2018 (c) Housing Price Inequality Index change rate between 2013 and 2018.

**Table 1 ijerph-18-09877-t001:** Summary of statistics.

Variable		Obs.	Mean	Std. Dev.	Min	Max
Dependent variable						
Suicidal impulse		24,392	0.031	0.173	0	1
Inequality attribute						
Housing price inequality (region)		170	0.238	0.029	0.160	0.352
Demographic attribute						
Gender (Female = 1, Male = 0)		24,392	0.590	0.492	0	1
Age		24,392	46.741	20.098	12	80
Married (Yes = 1, No = 0)		24,392	0.267	0.442	0	1
Number of family members (reference: 1 person)	2 persons	24,392	0.256	0.436	0	1
	3 persons	24,392	0.243	0.429	0	1
	4 persons	24,392	0.278	0.448	0	1
	5 persons	24,392	0.089	0.285	0	1
	over 6 persons	24,392	0.034	0.182	0	1
Education (reference: graduate elementary school)	Middle school	24,392	0.143	0.350	0	1
	High school	24,392	0.284	0.451	0	1
	University	24,392	0.261	0.439	0	1
Total income (10,000 KRW *)		24,392	3029	3707	17	18,000
Health condition attribute						
Physical health		24,392	3.093	0.928	1	5
Mental health		24,392	2.750	0.779	1	4

* KRW is roughly equivalent to USD 1.1 in the 2020 exchange rate.

**Table 2 ijerph-18-09877-t002:** Results for regression analysis between suicidal impulse and housing price inequality.

Variable	All Period (10 Years)			Recent Period (5 Years)		
	Model 1	Model 2	Model 3	Model 4	Model 5	Model 6
	O.R.(S.E.)	O.R.(S.E.)	O.R.(S.E.)	O.R.(S.E.)	O.R.(S.E.)	O.R.(S.E.)
Cons	0.019 *** (0.002)	0.000 *** (0.000)	0.000 *** (0.000)	0.019 *** (0.002)	0.000 *** (0.000)	0.000 *** (0.000)
Inequality attribute						
Housing price inequality		1.664 ** (0.428)	1.946 *** (0.439)		7.7297 *** (3.641)	5.355 *** (3.300)
Demographic attribute						
Gender (Female = 1, Male=0)			0.859 * (0.071)		1.039 (0.112)	
Age			0.998 (0.004)		1.007 (0.005)	
Married (Yes = 1, No = 0)			1.271 (0.187)		1.789 *** (0.352)	
Number of family members			0.939 * (0.035)		0.868 *** (0.046)	
Education			0.748 *** (0.030)		0.731 *** (0.038)	
Total income (log)			0.901 *** (0.024)		0.950 (0.034)	
Health condition attribute						
Physical health			0.759*** (0.034)		0.712 *** (0.044)	
Mental health			0.387 *** (0.019)		0.292 *** (0.019)	
*N*	24,392	24,392	24,392	18,771	18,771	18,771
LR test chi2	286.030 ***	235.590 ***	139.870 ***	223.650 ***	111.240 ***	123.630 ***
Wald chi2		3.920 ***	663.150 ***		18.850 ***	589.990 ***
Log likelihood	−3224.025	−3222.090	−2875.700	−1963.23	−1951.58	−1633.47
AIC	6452.052	6450.180	5777.405	3930.456	3911.154	3294.949
BIC	6468.256	6474.486	5882.731	3946.136	3942.515	3404.710
ICC (year | region)	0.164	0.158	0.108	0.205	0.129	0.152

Model 1, 4: unconditional mean model, Model 2, 3, 5, 6: random intercept model. Likelihood ratio (LR); intraclass correlation coefficients (ICCs); Akaike information criterion (AIC); Bayesian information criterion (BIC). *** *p* < 0.01, ** *p* < 0.05, * *p* < 0.1.

**Table 3 ijerph-18-09877-t003:** Results for regression analysis by income percentiles.

Variable	Income Level		
	Top 20%	Median 20%	Bottom 20%
	O.R.(S.E.)	O.R.(S.E.)	O.R.(S.E.)
Cons	0.000 *** (0.000)	0.000 *** (0.000)	0.003 *** (0.003)
Inequality attribute			
Housing price inequality	3.793 *** (1.567)	3.155 *** (1.272)	1.901 ** (0.553)
Demographic attribute			
Gender (Female = 1, Male = 0)	0.985 (0.237)	0.721 * (0.126)	0.722 ** (0.102)
Age	1.010 (0.013)	1.009 (0.008)	0.990 (0.006)
Married (Yes = 1, No = 0)	2.789 ** (1.431)	1.754 * (0.563)	1.007 (0.260)
Number of family members	0.960 (0.120)	1.009 (0.079)	0.943 (0.062)
Education	0.656 *** (0.071)	0.842 * (0.074)	0..824 ** (0.063)
Total income (log)	1.800 (0.684)	0.914 (0.161)	0.841 * (0.087)
Health condition attribute			
Physical health	0.723 ** (0.109)	0.754 *** (0.072)	0.730 *** (0.052)
Mental health	0.400 *** (0.063)	0.317 *** (0.035)	0.450 *** (0.036)
*N*	4880	4906	5187
LR test chi2	0.350	38.240 ***	37.170 ***
Wald chi2	84.600 ***	168.410 ***	174.740 ***
Log likelihood	−345.457	−608.405	−971.577
AIC	716.914	1242.810	1969.155
BIC	801.322	1327.287	2054.356
ICC (year | region)	0.021	0.175	0.116
Income range (10,000 KRW)	Upper 5880	2700~525	Under 200

Likelihood ratio (LR); intraclass correlation coefficients (ICCs); Akaike information criterion (AIC); Bayesian information criterion (BIC). *** *p* < 0.01, ** *p* < 0.05, * *p* < 0.1.

**Table 4 ijerph-18-09877-t004:** Results for regression analysis by regional features.

Variable	Model 1: Seoul-Gyeonggi		Model 2: Local Metropolitan Cities		Model 3: Region Type	
	Yes	Other	Yes	Other	City	Non-City
	O.R.(S.E.)	O.R.(S.E.)	O.R.(S.E.)	O.R.(S.E.)	O.R.(S.E.)	O.R.(S.E.)
Cons	0.000 *** (0.000)	0.000 *** (0.000)	0.000 *** (0.000)	0.000 *** (0.000)	0.000 *** (0.000)	0.014 *** (0.018)
Inequality attribute						
Housing price inequality	14.704 *** (14.667)	1.747 ** (0.406)	3.136 ** (1.580)	2.753 ** (1.212)	2.436 *** (0.573)	1.197 (0.467)
Demographic attribute						
Gender (Female = 1, Male = 0)	1.057 (0.157)	0.784 ** (0.077)	0.748 * (0.111)	0.806 (0.114)	0.932 (0.087)	0.645 ** (0.114)
Age	0.994 (0.006)	0.999 (0.004)	0.999 (0.006)	1.003 (0.076)	0.999 (0.004)	0.992 (0.008)
Married (Yes = 1, No = 0)	1.005 (0.253)	1.426 * (0.259)	1.405 (0.375)	1.424 (0.382)	1.429 ** (0.230)	0.617 (0.241)
Number of family members	0.937 (0.063)	0.938 (0.042)	0.928 (0.063)	0.956 (0.062)	0.937 (0.039)	0.942 (0.077)
Education	0.765 *** (0.052)	0.738 *** (0.036)	0.704 *** (0.050)	0.781 *** (0.057)	0.748 *** (0.032)	0.731 *** (0.076)
Total income (log)	0.948 (0.047)	0.884 *** (0.027)	0.890** (0.043)	0.903** (0.040)	0.897*** (0.027)	0.896 ** (0.049)
Health condition attribute						
Physical health	0.703 *** (0.060)	0.779 *** (0.041)	0.796 *** (0.063)	0.767 *** (0.058)	0.755 *** (0.038)	0.770 *** (0.074)
Mental health	0.360*** (0.033)	0.397*** (0.023)	0.373*** (0.033)	0.409*** (0.034)	0.361*** (0.020)	0.481 *** (0.050)
*N*	7031	17361	8863	7048	19533	4859
LR test chi2	38.900 ***	91.280 ***	44.790 ***	27.070 ***	89.250 ***	32.330 ***
Wald chi2	215.210 ***	457.130 ***	216.590 ***	214.090 ***	578.690 ***	101.520 ***
Log likelihood	−855.703	−2012.791	−887.826	−966.542	−2223.977	−645.957
AIC	1737.407	4051.583	1803.653	1961.085	4473.956	1317.915
BIC	1826.562	4152.489	1902.908	2057.132	4576.394	1402.266
ICC (year | region)	0.089	0.109	0.099	0.091	0.097	0.171

Model 1, 4: unconditional mean model, Model 2, 3, 5, 6: random intercept model. Likelihood ratio (LR); intraclass correlation coefficients (ICCs); Akaike information criterion (AIC); Bayesian information criterion (BIC). *** *p* < 0.01, ** *p* < 0.05, * *p* < 0.1.

## Data Availability

Suicidal ideation and individual characteristics variables were analyzed using data from the South Korea Centers for Disease Control and Prevention’s National Health and Nutrition Survey (https://knhanes.kdca.go.kr/knhanes/main.do). The housing price gap was based on the housing price transaction of the Ministry of Land, Infrastructure and Transport in South Korea (http://rtdown.molit.go.kr).

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
