# Peer review of "Precarious Suicide Behavior According to Housing Price Gap: A Case Study on South Korea"

_ijerph, 2021, doi:10.3390/ijerph18189877_

Round 1
Reviewer 1 Report
My one critical recommendation is that the channel(s) of he hypothesized causation must be clearly explained. Why should a particular type of inequality in a province, measured by a specific index, affect residents' suicidal thoughts and behavior?
- Is the hypothesized determinant of suicides (static) inequality, or (dynamic) volatility, or both?
- Is it the individuals' house value (or mortgage payments etc) that causes them to be depressed, or is it individuals' concern over inequality in community?
- How does land speculation, or even corruption play a role in this story? Both of these are policy concerns in Korea, and both have been associated with suicides.
- Perhaps the article should begin by mentioning some notable examples of suicides related to real estate, such as this: https://www.reuters.com/article/us-southkorea-property-idUSKBN2B505U
Technical comments
This paragraph appears to be full of contradictions, and must be rewritten:
"South Korea had flexible social mobility and fair economic distribution 38 during the industrial growth period in the 1970s and 1980s. However, since the middle of 39 the 20th century, the country has experienced intense inequality as a side effect of tremendous economic growth. The Gini coefficient of South Korea in 2017 was 0.36, which indi-41 cates inequality in the income of the society [6, 7]. South Korea's Gini coefficient is close 42 to that of the United States, the highest among the OECD countries"
The use of the Gini coefficient, instead of other indices more sensitive to dispersion in the tails, should be better motivated.
"the inequality index in the country has been worsening." - Evidence for this? Please clarify whether this is before/during COVID.
"price inequality variable used the Ministry of Land, Infra-63 structure, and Transport’s (MOLIT) housing transaction price data, and the suicide behavior variable used data from the National Health Impact Survey" - Are these nationally representative?
The literature review is well presented, but I wish there was more information on studies assessing the relationship between wealth inequality/volatility and suicides.
"In 2018 and 2013" - it's a bit strange to report the number reverse-chronologically.
"survey that measures over 500 health conditions, 184 such as health, nutrition, and personal characteristics for all citizens." - misleading, since this is a survey of 10,000, not a general census.
"Excluding 2014," - explain
"females accounted for 59 %" - why does a nationally representative survey have a 59% female share?
"annual average income is 3,092 million won." - this is a wrong figure
"In other words, in urban areas, housing price inequality, and suicidal thoughts had a negative correlation." - Should this say 'negative'?
Grammar must be significantly improved - proofreading by an English-proficient reader is an absolute must.
E.g.:
"If responses were choosing"
Author Response
Thank you very much for your detailed comment. Your comments and comments have greatly improved the quality of this paper. The research team has carefully reviewed your comments and reflected almost all of them into the text. Once again, thank you for leaving a comment on our article.
Point 1: You recommended that the pathways of the causal relationship between inequality and suicide postulated should clearly explain.
Response 1: In agreement with the reviewer's suggestion, this study explained the mechanism of the development of income inequality-relative deprivation-suicidal ideation in previous studies. In addition, we explained in Chapter 5 Discussion that a similar mechanism appeared in our study results (line 82-85, 382-384).
Point 2: You asked whether the inequality in this study is static or dynamic.
Response 2: Inequality in this study is a static inequality based on the results of statistical data for that year, and suicidal ideation caused by dynamic inequality (change in inequality) is the main issue of our research team's follow-up study. We explained this in the future research description section of this text (line 466-467).
Point 3: You asked whether individual house prices were the cause of depression or community inequality.
Response 3: The subject of this study is suicidal impulses caused by gaps in housing prices over high housing prices, and this issue is once again clearly stated in the text to reduce confusion among readers (line 64-67).
Point 4: You explained that land speculation or corruption plays a role, and it's good to mention articles recommended by reviewers.
Response 4: The article you recommended is a suicide related to the corruption of a public institution employee, and this case is a particular case and has nothing to do with the subject of this study. As a result, we did not feel the need to reflect on it in the manuscript.
Point 5: You suggested that the paragraphs(line 38-43) should rephrase because the content is contradictory.
Response 5: We agree with the reviewer's suggestion and rephrased the paragraph to be more explicit. For example, in the 1970s and 1980s, the paragraph was revised overall, changing the phrase "there was a fair economic distribution" to "inequality was not a social concern at the time" (line 39-46).
Point 6: You advised that it should append that the Gini coefficient covers all object distributions rather than just the tail.
Response 6: Based on the reviewer's opinion, we added, "The Gini coefficient covers all distributions, not just the two tails distribution of a group" (line 234-235).
Point 7: You advised that this study needs evidence that the inequality index is getting worse.
Response 7: Based on a study that analyzed inequality in South Korea intensively (Koo, 2021), we added the content of the study and cited references that the inequality index is worsening (line 49-50).
Point 8: You asked about the reliability of the data used in the study.
Response 8: The housing transaction price data used in this study is the national statistical data of the Ministry of Land, Infrastructure, and Transport. The National Health and Nutrition Survey data is the national statistical data of the Ministry of Health and Welfare, Centers for Disease Control and Prevention. We clarified the source of information in the article (line 212-214).
Point 9: You suggested that the content of wealth inequality should be reinforced while the literature review is well presented.
Response 9: Studies analyzing the relationship between wealth inequality and suicide are very insufficient. However, several previous studies have supplemented the relationship between wealth differences and suicide (line 135-142).
Point 10: You explained in the text that the order of 2018 and 2013 is strange.
Response 10: By the reviewer's comments, the paragraph has been modified as follows: We revised the order of 2018 and 2013 and described the current status of 2018 and the rate of change in 2018 compared to 2013 by reflecting others' opinions (line 195-205).
Point 11: You said the statement that the data surveyed all citizens was misleading.
Response 11: We removed "statistics for all citizens" from the manuscript in agreement with the reviewer's comments (line 217).
Point 12: You mentioned that there is a need for an explanation as to why 2014 was excluded from the study.
Response 12: In response to the reviewer's comments, we added a statement that the 2014 statistics which was not surveyed suicide-related questionnaires (line 255-256).
Point 13: You advised that it needs an explanation for why there are fewer males than females.
Response 13: Agreeing with the reviewer's point, we added the phrase that while the male ratio in the existing statistical survey was 45.3%, the ratio decreased to 41.0% by removing non-response (line 260-262).
Point 14: You pointed out that the average income in the data is too high.
Response 14: The previous number was incorrect, so we corrected the average income to 30.9 million (line 265).
Point 15: You suggested a review on whether it is correct to use negative in a phrase, “In other words, in urban areas, housing price inequality, and suicidal thoughts had a negative correlation.”
Response 15: Instead of using the term ‘negative,’ we rewritten the phrase as “suicidal impulse is high in regions with high housing price inequality.” (line 360-361).

Reviewer 2 Report
The present study refers to very important relations between mental health (suicide impuls) and socio-economis conditions of the region and population. This study analyzes the impact relationship between suicidal impulses and economic inequality in South Korea. Keeping in mind South Korean's statistics it's even more necessary to investigate the reasons of this issue.
Conducted study is interesting and definitely worth publishig. Both models and methods are well-described and the results are clearly presented. Moreover, all conclusions were justified and supported by the results. The authors awareness of the limitations of the research is worth mentioning as well.
It could be assumed, that people living in big cities are dealing with more menthal problems,, despite the higher level of income, which was confirmed in present study.
Presumably, the paper will attract a wide readership and the results provide an advance in current knowledge.
However, it's worth adding in the text, how the results correspond with the discussion about Easterlin paradox and according to the results (lines 296-301; 404-413; 450-465) should we treat dwellings in metropolitan/capital areas as a positional good (according to Hirsch)?
Please note, that the description of Figure 1 is doubled, whereas there is no adequate description of Figure 2.
Author Response
Thank you very much for your detailed comment. Your comments and comments have greatly improved the quality of this paper. The research team has carefully reviewed your comments and reflected all of them into the text. Once again, thank you for leaving a comment on our article.
Point 1: You suggested that a description should be included that is consistent with the discussion of the Easterlin paradox.
Response 1: Agreeing with the reviewer's opinion, I added Easterlin's paradox to explain that the influence of income decreases when the income level is above the top 20% (line 401-403). We think that the addition of the Easterlin paradox has enriched the meaning of this study. Thank you so much for making our research worthwhile.
Point 2: You pointed out that Figure 1 was described twice.
Response 2: Since the existing figure 2 was misrepresented as figure 1, the second figure 1 was corrected to figure 2.
Point 3: You pointed out that the explanation in Figure 2 is not appropriate.
Response 3: In agreement with the reviewer's opinion, the explanation in Figure 2 has been rearranged to explain the current situation in 2018 and the rate of change from 2013 to 2018 (line 195-205).
